# Hunting regulation favors slow life histories in a large carnivore

Joanie Van de Walle[1,2], Gabriel Pigeon[1], Andreas Zedrosser[3,4], Jon E. Swenson[5,6] & Fanie Pelletier[1,2]

As an important extrinsic source of mortality, harvest should select for fast reproduction and accelerated life histories. However, if vulnerability to harvest depends upon female reproductive status, patterns of selectivity could diverge and favor alternative reproductive behaviors. Here, using more than 20 years of detailed data on survival and reproduction in a hunted large carnivore population, we show that protecting females with dependent young, a widespread hunting regulation, provides a survival benefit to females providing longer maternal care. This survival gain compensates for the females' reduced reproductive output, especially at high hunting pressure, where the fitness benefit of prolonged periods of maternal care outweighs that of shorter maternal care. Our study shows that hunting regulation can indirectly promote slower life histories by modulating the fitness benefit of maternal care tactics. We provide empirical evidence that harvest regulation can induce artificial selection on female life history traits and affect demographic processes.

[1] Département de biologie, Université de Sherbrooke, Sherbrooke, QC J1K 2R1, Canada. [2] Centre for Northern Studies, Université Laval, Quebec, QC G1V 0A6, Canada. [3] Department of Natural Sciences and Environmental Health, University of Southeast Norway, NO-3800 Bø i Telemark, Norway. [4] Institute of Wildlife Biology and Game Management, University of Natural Resources and Life Sciences, A-1180 Vienna, Austria. [5] Faculty of Environmental Sciences and Natural Resource Management, Norwegian University of Life Sciences, NO-1432 Ås, Norway. [6] Norwegian Institute for Nature Research, NO-7485 Trondheim, Norway. Correspondence and requests for materials should be addressed to J. Van de Walle. (email: joanie.van.de.walle@usherbrooke.ca) or to A.Z. (email: andreas.zedrosser@usn.no)

The strong potential for human harvest to alter ecological and evolutionary processes has been recognized in marine systems, where harvest pressure has direct and indirect effects on both fish yield and economic revenue[1–4]. Size-selective harvest of larger fish favors maturation at smaller sizes and reduces yield[5]. In terrestrial systems, trophy hunting can artificially increase mortality of individuals with large horns, tusks, or antlers, which can induce selective pressures and lead to evolutionary changes in heritable morphological traits that cannot be quickly reversed by natural selection[6–8]. Although it is easily conceivable that size-selective harvest regimes can alter patterns of natural selection and cause demographic changes in wildlife populations, such effects are usually less expected and thereby less studied in harvest regimes that are not size selective[9]. However, theoretical models show that size-independent harvest can also induce selective pressures on life history traits[10–13], and recent empirical work has documented harvest selectivity for age, sex, and behavioral traits in the wild[14–16]. Despite its importance for management, we still know very little about the consequences of such selectivity on population processes.

Fitness is maximized by allocating resources to survival or reproduction and the function over which allocation will be biased depends on levels of extrinsic mortality[17]. Under high extrinsic mortality, fitness can be optimized by higher investment in reproduction, leading to selection for faster life histories[5,13], similar to what would be expected under natural selection. This acceleration in the pace of life has been documented in several harvested populations, which suggests that harvest should lead to r-selection[2,13,16,18]. Moreover, even in the absence of any harvest preferences, individuals can differ in vulnerability to harvest, depending on behavior, harvest methods, and regulations[16,19–21]. Indeed, in several sport hunting systems, the killing of females with dependent offspring is either illegal, discouraged, or avoided by hunters to protect the female segment of the population or because of the potentially lowered survival of orphaned offspring that can cause ethical, as well as demographic, issues[22–25]. In such systems, reproducing females are less vulnerable to hunting and thus should enjoy an artificial selective advantage that is accentuated with increasing hunting pressure. This type of harvest selectivity could promote longer periods of mother-offspring associations and slower life histories, with potential consequences for population dynamics. Although the potential selective and demographic effects of the protection of females based on reproductive status have already been acknowledged[20,22–27], these effects have rarely been quantitatively assessed using empirical data[16].

Here, we test whether a hunting regulation that prohibits the killing of females with dependent offspring can induce selectivity on female reproductive tactics at the individual level and evaluate the effect of such selectivity on population processes. We use more than 20 years of exceptionally detailed individual-based data on survival and reproduction in a heavily hunted population of brown bears (*Ursus arctos*) in Sweden[28], where two distinct maternal care tactics have been documented[29]. We start by documenting the temporal trend in the duration of maternal care and contrasting survival probabilities between females providing either short (1.5-year tactic) or long (2.5-year tactic) maternal care. Longer maternal care entails a loss of reproductive opportunities in species where breeding is not resumed until current offspring are weaned[30]. Therefore, we compare two demographically and evolutionary meaningful proxies of fitness[31] that integrate information on survival and reproduction, i.e., asymptotic population growth rate ($\lambda$; the annual per capita rate of population increase[32]) and net reproductive rate, $R_0$ (number of females an individual is expected to produce over its lifetime[31,33]), between the two maternal care tactics to quantify the difference in

fitness between these tactics. Finally, we complement this analysis by evaluating the fitness pay-off of each maternal care tactic under various plausible scenarios of hunting pressure to determine if hunting can drive the relative occurrence of maternal care tactics in the population.

We show that being in a family group and providing longer maternal care results in a survival advantage for both adult females and dependent offspring. This survival advantage compensates for a reduction in reproductive output for females providing longer maternal care. As a result, both maternal care tactics have similar fitness values on average, but as hunting pressure increases, longer maternal care yields higher fitness returns. Protecting females with dependent young has therefore the potential to induce selectivity towards a lengthening of maternal care in the Scandinavian brown bear, with consequences for population generation time and age structure.

## Results

**Occurrence of maternal care tactics in the population**. We found that from 1987 to 2015, the odds of a litter being weaned after 2.5 years of maternal care increased by a factor 1.17 (95% CI = [1.07, 1.29]) annually in the population ($n = 164$ litters from 62 individual females). The first documented litter in our study population that was raised with the 2.5-year tactic was born in 1993 and weaned in 1995, and since then 24.8% of the litters have received 2.5 years of maternal care; the rest received maternal care for 1.5 years (Fig. 1).

**Protective effect of maternal care**. Adult ($\geq 4$ years-old (y.o.)) and yearling female brown bears in a family group during the hunting season have a survival advantage compared to when solitary (Fig. 2a). For adult females, being solitary during the hunting season significantly reduces survival probability (odds-ratio: 0.30, 95% CI = [0.16, 0.57]). The annual finite mortality rate of solitary adult females was 0.16 (95% CI = [0.12, 0.20], $n = 407$ bear-years) and the hunting-induced annual finite mortality rate was 0.14 (95% CI = [0.10, 0.17], 55 cases). In contrast, the annual finite mortality rate of adult females accompanied by dependent offspring (i.e. member of a family group) was only 0.06 (95% CI = [0.03, 0.10], $n = 207$ bear-years, Fig. 2b) and the hunting-induced annual finite mortality rate was 0.04 (95% CI = [0.02, 0.07], 9 cases during the study period 1993–2015, where a hunter accidentally shot the female before observing the dependent offspring). Therefore, the odds of dying from hunting were 3.91 times higher for solitary females compared to adult females accompanied by dependent offspring.

**Tactic-based demographic analyses**. Focusing from 1993 to 2015, when both maternal care tactics coexisted in the population, we found that individual females had consistent durations of maternal care (repeatability = 0.33, 95% CI = [0.11, 0.47]), suggesting that the population includes two distinct behavioral tactics regarding maternal care. Most females (62.5%) consistently provided maternal care for 1.5 or 2.5 years exclusively, whereas the rest could alternate between the 1.5-year and 2.5-year tactics, but using one of the two tactics more frequently. Females that we considered to be consistent in their maternal care tactic were not older on average (mean = 10.88 y.o., 95% CI = [9.65, 12.11], $n = 25$) than more flexible females (mean = 12.07 y.o., 95% CI = [10.34, 13.81]; $t_{38} = -1.12$, $P = 0.27$, $n = 15$). Primiparous females had a similar probability of using the 1.5-year tactic compared to multiparous females (odds-ratio = 0.46, 95% CI = [0.05, 4.67], $n = 123$), suggesting that maternal experience was not the main factor explaining differences in maternal care tactics. Therefore, we separated adult females according to the

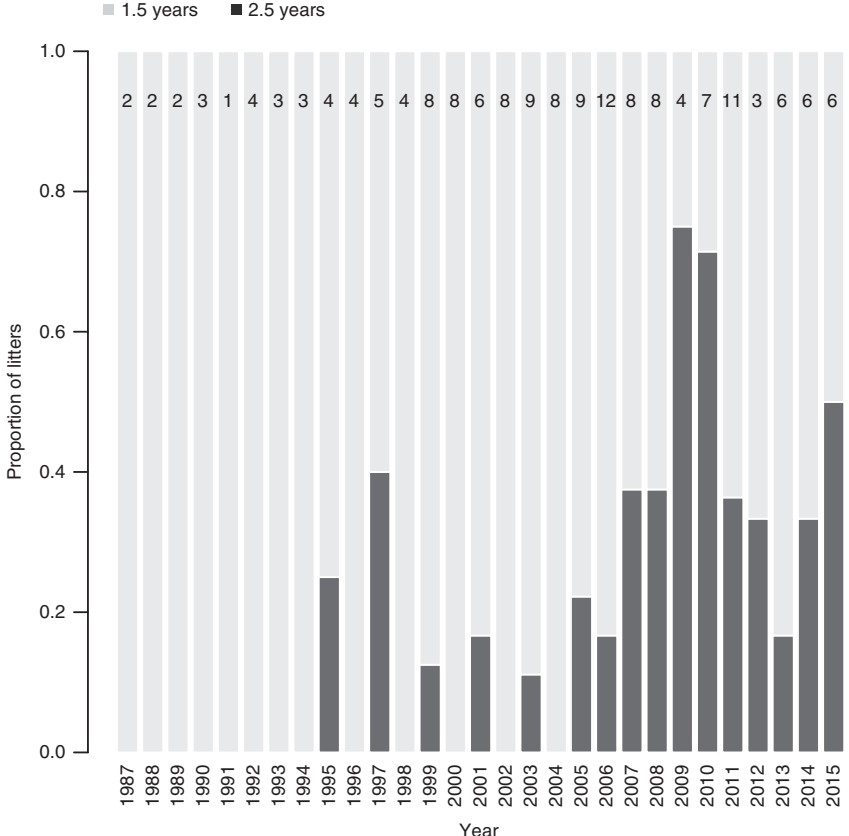

**Fig. 1** Temporal change in the duration of brown bear maternal care. Proportion of successfully weaned brown bear litters ($n = 164$ from 62 individual females) that had received 1.5 years and 2.5 years of maternal care in south-central Sweden from 1987 to 2015. Sample size for each year is indicated on top of each bar

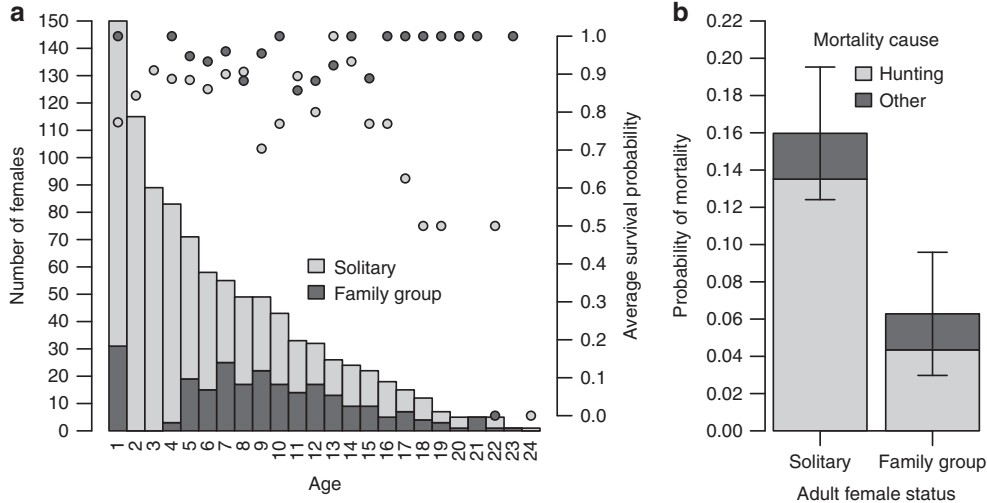

**Fig. 2** Protective effect of hunting regulation for brown bear females. **a** Age-specific number (bars, left axis) and average survival probability (dots, right axis) of female brown bears according to their status (solitary: light gray; member of a family group: dark gray) in south-central Sweden from 1993 to 2015. All cubs-of-the-year (age = 0) are dependent upon their mother and are protected from hunting by regulation, however, they are not represented in this figure, because their sex could not be determined. **b** Adult female probability of mortality (average and 95% CI) from hunting and other causes according to their reproductive status (solitary, $n = 407$; member of a family group, $n = 207$)

duration of maternal care they provided and young females (≤3 y. o.) according to the duration of maternal care they had received as cubs. We estimated tactic- and age-specific demographic rates (Supplementary Table 1), survival, and recruitment as the number of yearling daughters produced per year per female, following

the previously described age structure (Supplementary Fig. 1) in this population[34]. We found that age class and maternal care tactic were both important factors explaining variation in survival (Table 1). Prime-age females, aged 4–8 y.o., enjoyed the highest survival independently of maternal care tactic. Overall, the odds

**Table 1 Parameter estimates from final statistical models comparing tactic- and age class-specific demographic rates for female brown bears in south-central Sweden from 1993 to 2015**

|  | Coefficient | Lower 95% CI | Upper 95% CI | z-value | P-value |
|---|---|---|---|---|---|
| **Survival** |  |  |  |  |  |
| Intercept | 1.430 | 1.015 | 1.887 | 6.489 | <0.0001 |
| 2.5-year tactic | 0.761 | 0.214 | 1.365 | 2.610 | 0.009 |
| Age class 2 y.o.[a] | 0.238 | −0.434 | 0.934 | 0.685 | 0.494 |
| Age class 3 y.o. | 0.932 | 0.099 | 1.887 | 2.071 | 0.038 |
| Age class 4–8 y.o. | 2.124 | 1.219 | 3.230 | 4.229 | <0.0001 |
| Age class 9+ y.o. | 0.224 | −0.335 | 0.769 | 0.798 | 0.425 |
|  | Variables removed[b]: | | | | |
|  | Tactic*Age class ($X^2 = 7.920$, $P = 0.095$) | | | | |
| **Recruitment** |  |  |  |  |  |
| Intercept | −0.923 | −1.154 | −0.697 | −7.924 | <0.0001 |
| 2.5-year tactic | −0.425 | −0.860 | −0.003 | −1.948 | 0.051 |
|  | Variables removed: | | | | |
|  | Tactic*Age class ($X^2 = 1.935$, $P = 0.164$) | | | | |
|  | Age class ($X^2 = 0.969$, $P = 0.325$) | | | | |

Parameters come from binomial and negative binomial models of survival probability and recruitment rate (i.e., the number of yearling daughters produced per female per year), respectively. Variables were removed if their inclusion did not improve model fit according to likelihood ratio tests. Results are presented on their transformed scale to show statistical significance
[a]y.o. = years-old
[b]the star is used to represent interactions

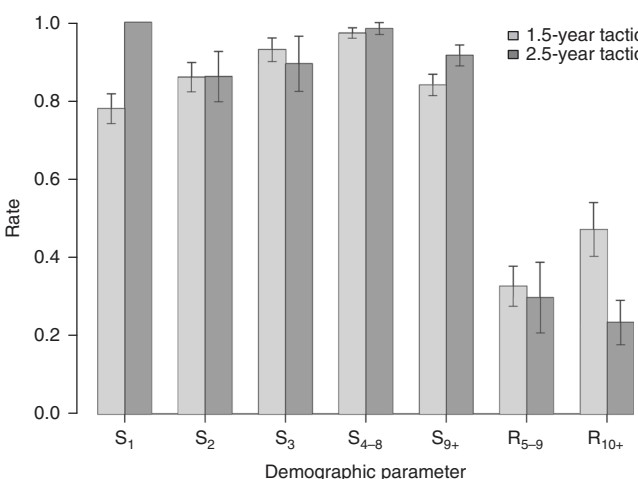

**Fig. 3** Tactic and age class-specific demographic parameters for female brown bears. Empirical values come from longitudinal data on female brown bears monitored in south-central Sweden from 1993 to 2015. Bars represent 95% confidence intervals. S is annual finite survival rate and R is recruitment, i.e., the number of yearling daughters produced per female per year, $S_1$ is survival of yearlings, $S_2$ is survival of 2 year-olds (y.o.), $S_3$ is survival of 3 y.o., $S_{4-8}$ is survival of 4–8 y.o., $S_{9+}$ is survival of 9 y.o. and older, $R_{5-9}$ is recruitment of 5–9 y.o., and $R_{10+}$ is recruitment of 10 y.o. and older. The descriptive statistics presented on the figure and sample sizes can be found in Supplementary Table 1

**Table 2 Bootstrapped model-based predictions (back-transformed on the original scale) of tactic- and age class-specific demographic rates for female brown bears in south-central Sweden from 1993 to 2015**

|  | 1.5-year tactic | | | 2.5-year tactic | | |
|---|---|---|---|---|---|---|
|  | Mean estimate | 95% CI | | Mean estimate | 95% CI | |
|  |  | Lower | Upper |  | Lower | Upper |
| **Survival** |  |  |  |  |  |  |
| $S_1$ | 0.809 | 0.739 | 0.874 | 0.903 | 0.833 | 0.951 |
| $S_2$ | 0.844 | 0.767 | 0.911 | 0.922 | 0.860 | 0.966 |
| $S_3$ | 0.916 | 0.851 | 0.974 | 0.961 | 0.914 | 0.988 |
| $S_{4-8}$ | 0.973 | 0.947 | 0.994 | 0.988 | 0.971 | 0.997 |
| $S_{9+}$ | 0.841 | 0.789 | 0.889 | 0.920 | 0.874 | 0.957 |
| **Recruitment** |  |  |  |  |  |  |
| $R_{5-9}$[a] | 0.384 | 0.296 | 0.482 | 0.251 | 0.167 | 0.351 |
| $R_{10+}$[b] | 0.384 | 0.296 | 0.482 | 0.251 | 0.167 | 0.351 |

Model predictions were bootstrapped 10,000 times to generate average estimates and 95% confidence intervals. S is survival and R is recruitment, i.e., the number of yearling daughters produced per female per year, $S_1$ survival of yearlings, $S_2$ survival of 2 year-olds (y.o.), $S_3$ survival of 3 y.o., $S_{4-8}$ survival of 4–8 y.o., $S_{9+}$ survival of 9 y.o. and older, $R_{5-9}$ recruitment of 5–9 y.o., and $R_{10+}$ recruitment of 10 y.o. and older.
[a] Because reproductive rates are represented by fecundities (Fecundity$_t$ = Survival$_{(t \to t+1)}$ × Recruitment$_{t+1}$) in the tactic-specific matrix models, recruitment was estimated for age classes 5–9 and 10+ years to follow age classes for survival, and because 5 years is the youngest age at which females may start producing yearlings in our study population.
[b]Age class did not significantly affect recruitment rate, thus, age class 5–9 and 10+ years were assigned a similar recruitment value in matrix models

of surviving were 2.14 higher (95% CI = [1.21, 3.79]) for females using the 2.5-year tactic and the difference in predicted survival probabilities, all age classes combined, was 5.1% between the two maternal care tactics (average predicted survival probability for the 2.5-year tactic: 0.95, average predicted survival probability for the 1.5-year tactic: 0.90). This clearly shows the survival benefit of longer periods of maternal care for both young and adult females. The interaction between tactic and age class did not improve model fit significantly (Table 1), suggesting that the management regulation has a protective effect for all age classes. Increased survival probability for the 2.5-year tactic, however, was most apparent in yearling and adult females (Fig. 3), because only those

age classes benefit from the protective regulation. In comparison, survival probabilities were similar between maternal care tactics for 2- and 3-year-olds, because at those ages all females are independent from their mothers and are thus equally vulnerable to hunting. Survival analyses produced another key result; all female yearlings raised with the 2.5-year tactic survived when accompanying their mother in their second year, whereas independent yearling females had a lower survival probability of 0.78 ± 0.04 (Supplementary Table 1). However, since adult females cannot mate prior to weaning their offspring, they will have fewer reproductive opportunities when using the 2.5-year tactic. Indeed, we found that using the 2.5-year maternal care tactic reduces

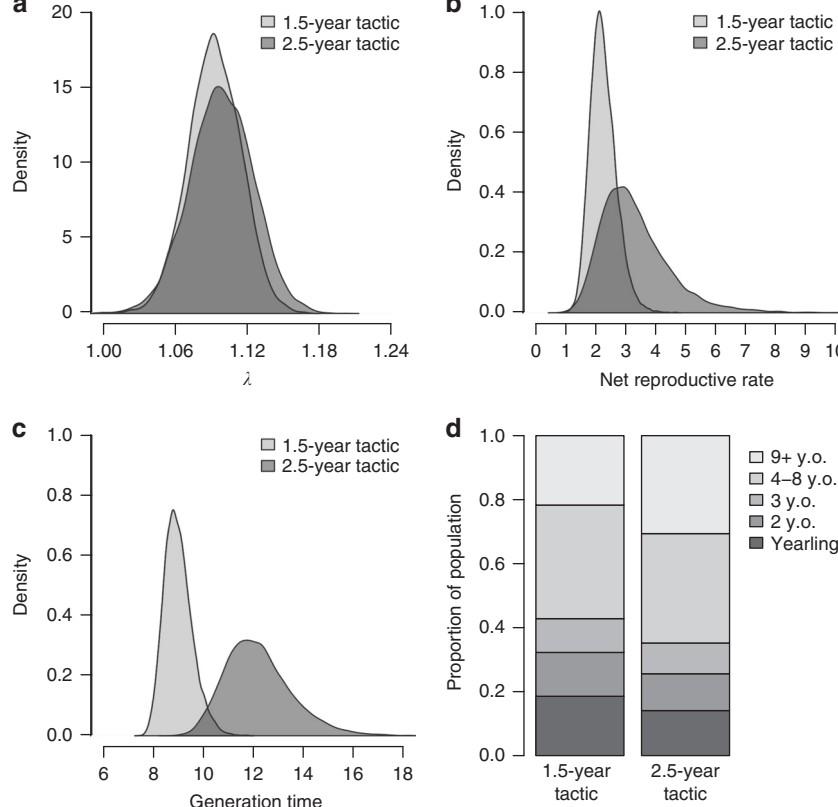

**Fig. 4** Contrasted demographic parameters between brown bear maternal care tactics. **a** Asymptotic population growth rates, $\lambda$, **b** net reproductive rates, $R_0$, **c** generation times, $T$, and **d** stable age structures were extracted from tactic-based population matrices using bootstrapped predictions of age class-specific survival probabilities and recruitment (10,000 predictions per demographic parameter per age class, yielding 10,000 different population matrices). Table 2 shows the bootstrapped model predictions for age-specific survival probabilities and recruitment rates that were used in the population matrices to generate the demographic parameters for each tactic

adult females recruitment (Table 1), which was 34.6% lower (yearly recruitment = 0.25) compared to the 1.5-year tactic (yearly recruitment = 0.38; Table 2).

The two maternal care tactics have contrasting costs and benefits for survival and reproduction, the two main components of individual fitness. That raises a crucial question: which tactic yields the highest overall fitness? Incorporating model predictions for tactic- and age-specific survival probabilities and recruitment (Table 2) into tactic-based population matrices yielded an asymptotic population growth rate ($\lambda$) of 1.09 (95% CI = [1.05, 1.14]) for the 1.5-year tactic, and of 1.10 (95% CI = [1.05, 1.15]) for the 2.5-year tactic. The net reproductive rate tended to be lower for the 1.5-year tactic, 2.26 (95% CI = [1.54, 3.22] compared to the 2.5-year tactic, 3.32 (95% CI = [1.68, 6.23]). However, both tactics yielded overlapping confidence intervals and bootstrapped distributions of $\lambda$ and $R_0$ (Fig. 4a,b), suggesting similar fitness over the study period.

Although $\lambda$ and $R_0$ were similar between maternal care tactics, other population processes may be affected by a switch in maternal care tactics in the population. First, generation time, $T$, (i.e., the time required for the population to be multiplied by its net reproductive rate, $R_0$[33]) would be lengthened by about 3 years, should the population be comprised of only females using the 2.5-year tactic (mean generation time 1.5-year tactic: 8.95 years, 95% CI = [8.08, 10.27]; mean generation time 2.5-year tactic: 12.05 years (95% CI = [10.03, 15.24]). Although the bootstrapped distributions of generation times slightly overlapped (Fig. 4c), the simulated generation times for the 2.5-year tactic had a 99.5% probability of being higher than for the 1.5-year tactic. A hunting

regulation favoring the 2.5-year tactic is thus most likely to promote slower life histories in this population. Second, stable age structures extracted from the tactic-specific matrix models contained more adult females within the 2.5-year tactic compared to the 1.5-year tactic (64.8% vs. 57.2%; Fig. 4d).

**Differential effects of hunting between tactics.** We found that the fitness benefit of prolonged maternal care increases with increasing levels of hunting pressure. Over the study period, hunting pressure varied from 0% to 34% (Supplementary Fig. 2). By allowing survival components to vary as a function of observed hunting pressure in the tactic-specific population matrices, we show that, whereas $\lambda$ decreases steadily with increasing hunting pressure for the 1.5-year tactic, it remains relatively unaffected by hunting pressure for the 2.5-year tactic (Fig. 5). This suggests that hunting may affect the relative occurrence of reproductive tactics in populations subject to a hunting regime that is selective regarding female reproductive status.

## Discussion

Humans as predators are a dominant agent of mortality in wildlife populations[1] imposing a selective landscape that vary both in its strength and phenotypic targets[14,15,18,20,35]. Here, we show that a hunting regulation based on female reproductive status can improve the survival prospects of female brown bears that provide longer maternal care, thereby promoting slow life histories, with consequences for population processes.

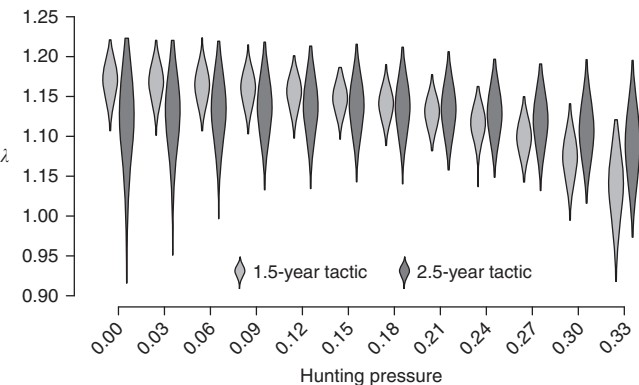

**Fig. 5** Hunting-induced change in the occurrence of brown bear maternal care tactics. Predicted effect of hunting pressure (number of marked bears that were shot in a given year divided by the number of marked bears available for hunting that same year) on the asymptotic population growth rate, $\lambda$, for each maternal care tactic using longitudinal data on female brown bears from 1993 to 2015 in south-central Sweden. The violins represent the density plots of lambda (1000 iterations) at each hunting pressure simulated. Estimates of the effect of hunting pressure on tactic- and age class-specific survival probabilities are given in Supplementary Table 2

Over the study period, both maternal care tactics yielded overlapping asymptotic population growth. This might appear surprising, considering that differences in recruitment (35.2%) are larger than differences in survival probability (5.1%) between the two tactics. However, because the elasticities of demographic rates in long-lived mammals are usually larger for survival than for reproduction[34,36,37], the gain in survival for females using the 2.5-year tactic may explain the similar fitness between the two tactics. The artificial gain in survival through prolonged maternal care due to legal protection from hunting can compensate for reduced reproductive output in hunting systems where family groups are protected. In a Swedish population of moose (*Alces alces*), where calves must be shot first before hunters are allowed to kill the mother, regulations artificially reduce the costs of reproduction by lowering the mortality of females with dependent calves, because calves act as a shield or because hunters prefer to harvest nonlactating females[26]. Moreover, because it integrates the survival probabilities of both adults and offspring, the lack of differences in $\lambda$ between the two tactics may also be explained partially by differences in offspring survival. Indeed, longer maternal care may increase offspring survival probabilities through higher energy intake and additional protection from other causes of mortality that independent offspring may face. In leopards (*Panthera pardus*), longer maternal care in adverse years acts as a buffer against prey scarcity, which compensates a female's loss of reproductive opportunities[30]. Here, we show that the survival gain due to management regulations can compensate for reduced recruitment, but also promote alternative maternal care tactics in harvested populations.

Over the study period, we expected that shorter periods of maternal care would be more advantageous, because it provides females with more reproductive opportunities and because the energetic costs associated with maternal care are high in mammals[38]. Also, strong historical persecution, regardless of reproductive status, has favored high investment in reproduction in the Scandinavian brown bear[39]. The resulting short interbirth intervals have been a key element for the rapid recovery of the population, that increased from about 130 individuals in the 1930s to around 2800 in 2013, following the implementation of protective measures[40]. Rapid reproduction is especially advantageous in expanding populations below carrying capacity, but as population density increases investment into survival becomes more advantageous[41]. Compared to its historic value, our study population is currently experiencing a reduction in its growth rate[34,40], which could explain why fast reproduction may be less advantageous today, even with relatively lower levels of hunting pressure. In addition, it is possible that intraspecific killing of yearlings, an important cause of mortality for female yearlings[42], may be higher today. Indeed, because surviving male bears reorganize their homes ranges after the death of a nearby male[43], hunting can promote spatial reorganization, which, combined with higher numbers of bears in the population in general, leads to higher probabilities of deadly encounters between young bears and adult males[42,44]. By staying with their mother an additional year, yearlings not only gain protection against hunting, but also against their other main cause of mortality; intraspecific killing[28,42]. The survival benefit of prolonged maternal care did not carry over, however, as the survival probabilities were similar between tactics for age classes 2 and 3 y.o., which is in line with other studies in birds and mammals that have found contradicting results from carry over effects of the duration of maternal care[30,45]. However, in the context where maternal care acts as a shield against hunting, prolonged maternal care has the additional advantage of reducing the period of vulnerability between weaning/independence and first reproduction, when further protection from hunting can be obtained.

The relative occurrence of the 2.5-year maternal care tactic has increased in south-central Sweden since 1987. Before 1995, the 2.5-year tactic was unobserved, although this could be due to the lower number of litters monitored in the early stages of the study. Up until 2005, when a larger number of litters has been monitored ($n = 84$), the 2.5-year tactic remained relatively rare and accounted for only 7.1% of the litters weaned, whereas this percentage increased to 36.3% from 2005 to 2015 ($n = 80$). Offspring mass and resource availability are often cited as the most important drivers of the duration of maternal care, with smaller offspring usually being cared for longer[29,46]. It is possible that a potential reduction in resources availability could have forced females to extend their maternal care in recent years. However, the abundance of bilberry (*Vaccinium myrtillus*), the most important food resource driving variations in body mass and reproductive success of Scandinavian brown bear females, has shown variations over the last ten years in the study area, but it did not decline[47]. Also, these bears can switch to alternative food items, such as crowberry (*Empetrum* spp.), when bilberry abundance is low, which suggests that they are less vulnerable to food shortage[48]. Alternatively, an increase in hunting quotas[40] and hunting pressure since 1993 (Supplementary Fig. 2) may have disproportionally removed fast-reproducing females, thereby artificially selecting for females that provide longer periods of maternal care in the population. Indeed, we show that the fitness pay-off of each reproductive tactic depends on harvest intensity, with lower hunting pressure selecting for shorter maternal care and higher hunting pressure selecting for longer maternal care. In the side-blotched lizard (*Uta stansburiana*), natural selection favors alternative female reproductive strategies when population density cycles, with slower life histories being selected at high densities[41]. In our study population, although changes in hunting pressure could correlate with density effects, we did not observe a similar trend between the fitness of tactics as a function of population density (Supplementary Fig. 3). A more likely explanation for the increasing gap in $\lambda$ between the tactics with increasing hunting pressure is the overall stronger negative effect of hunting on the survival of females using the 1.5-year tactic, especially of adults and yearlings, compared to females using the 2.5-year tactic (Supplementary Table 2). Prolonged maternal care

provides a buffer against high hunting pressure, as it protects adult females, which are the most productive segment of the population[37], as well as yearlings, which are the most vulnerable individuals[28]. This implies that the relative frequency of female reproductive tactics may alternate over time in the population depending on the level of human exploitation, as previously suggested by a mathematical modeling based on red deer (*Cervus elaphus*) life histories[11]. Over our 22-year study period and under intermediate values of hunting pressure, both tactics showed similar fitness, which suggests that the two phenotypes will likely be maintained in the population, as it is the case in many other brown bear populations[49,50]. However, if levels of hunting pressure continue to be high or increase, our study suggests that the relative occurrence of the 2.5-year tactic will increase.

Despite a similar asymptotic population growth between the 1.5-year and the 2.5-year tactics, a change in maternal care tactic can affect other population processes. Using simulations, it has been shown in the alpine chamois (*Rupicapra rupicapra*) that selective harvest of nonlactating females may affect age-specific mortality and population age structure, especially at high harvest rates[25]. Here, using empirical data on tactic-specific demographic rates, we show that two hypothetical brown bear populations, one consisting of females using the 1.5-year tactic and the other consisting of females using the 2.5-year tactic, would show different age structures, but without detectable consequences for population growth. This suggests that interactions between hunting pressure and female reproductive tactics likely affect population processes indirectly through other demographically important variables. A shift in population age structure could affect sensitivities of demographic rates and patterns of evolutionary dynamics, which would be missed by focusing on population growth alone[51]. For example, a shift in population age structure towards the adult female segment of the population may further divert the hunt towards solitary adult females and select for even longer maternal care. Moreover, because adult female survival has the greatest elasticities, such a population would be highly sensitive to a removal of the legal protection of family groups. Also, in such a population, harvest-induced evolutionary changes would take even longer to reverse, as generation time would be lengthened.

In this study, we used detailed empirical data from individual-based survival and reproduction to show that a hunting regime protecting females based on their reproductive status can induce selective pressures on female life history traits. The protection of (or unwillingness to kill) females with young is widespread among hunting systems, with examples from a vast range of game species[20,24,25,52,53]. Because of the survival advantage gained by females accompanied by dependent young, there is a great potential to observe selectivity based on female reproductive status in several other hunting systems. This selectivity could be towards longer periods of maternal care in populations where this trait varies[30,46,54–56], or may favor higher investment in reproduction early and late in life. This management-induced selectivity should not be overlooked, as it acts on female life history traits, which are by definition the very drivers of female fitness and population dynamics. Interestingly, however, a switch in female maternal care tactic had no effect on population growth in our system. Studies investigating indirect effects of hunting by monitoring changes in population growth alone are thus likely to miss important changes in female life histories and demography. Such changes may also impact hunters through a reduction in the availability of adult females and an unintentional, as well as potentially undesirable, hunting bias towards subadult females and males. Understanding how hunting and management regulations interact with animal life histories to affect population

processes is thus of great ecological, management, and conservation importance.

## Methods

**Study system and species.** The study area is located in Dalarna and Gävleborg counties, south-central Sweden (approximately 61° N, 15° E). The area encompasses 13,000 km² of rolling landscape dominated by heavily managed Scots pine (*Pinus sylvestris*) and Norway spruce (*Picea abies*) forests. Hunting is the most important source of mortality for adult Scandinavian brown bears[28]. Bears are hunted throughout the study area; the bear hunting season starts in late August and lasts until mid-October or until the regional quota has been filled. No specific license is required to hunt bears in Sweden. Hunting is allowed for anyone possessing hunting rights in a hunting area and a weapon legal for big game hunting[57]. Hunters can kill any solitary bear; however, bears in family groups are protected by law. All successful brown bear hunters must notify an officially appointed inspector on the day of the kill.

Female Scandinavian brown bears give birth to 1–4 cubs in January while in their den[58]. Females provide maternal care for either 1.5 or 2.5 years and family break-up generally occurs in spring soon after den emergence and before or during the mating season in May and June[59]. The mean inter-birth interval is 2.4 years in our study population, suggesting that most brown bear females that have separated from their offspring mate again in the spring and give birth to a new litter the following winter in our study population[50]. After families separate, all members may be hunted legally in the fall. Therefore, assuming no pre-weaning losses, a female using the 2.5-year tactic would be available for hunting in 1 of 3 years and her offspring could not be legally harvested as yearlings. A female using the 1.5-year tactic would be available for hunting every second year, and her yearlings would be vulnerable to harvest as well. We used data on brown bear females aged 1–24 y.o. from a long-term research project in south-central Sweden, where the recapture probability of marked females is almost 100%[28]. We focused our demographic analyses from 1993 to 2015, when the two maternal care tactics coexisted in the population. It was not possible to record data blind because our study involved marked individuals from a longitudinal long-term monitoring program and no randomization was used. Our handling of study animals was approved by the appropriate authorities and ethical committee: the Swedish Board of Agriculture (no. 35-846/03, 31-7885/07, 31-11102/12), the Uppsala Ethical Committee on Animal Experiments (no. C40/3, C47/9, C7/12), and the Swedish Environmental Protection Agency (no. 412-7327-09 Nv). We used R version 3.3.0 for all demographic and statistical analyses[60].

**Occurrence of maternal care tactics in the population.** To determine whether the relative occurrence of the 2.5-year tactic increased in the population over time, we used a binomial mixed effects model, with the probability that a litter received 2.5 years of maternal care as the response variable ("1" = received 2.5 years of maternal care, "0" = received 1.5 years of maternal care), and year as an explanatory variable. We also included female identity as a random factor to account for multiple observations of the same mother over time.

**Protective effect of maternal care.** To determine whether being in a family group provided a survival benefit to adult females, we used a generalized linear mixed effects model and tested for the effect of female status ("in a family group" or "solitary") during the hunting season on female survival probability ("1" = survival, "0" = mortality), while controlling for female age (fixed effect) and year (random effect). For this analysis, we used all females with known reproductive status (n = 614 bear-years).

**Tactic-based demographic analyses.** To contrast fitness components and resulting demographic characteristics between maternal care tactics, we first classified females according to their tactic. Subadult females (≤3 y.o.) were classified according to the duration of maternal care they had received. Adult females (≥4 y.o.) were classified according to the maternal care tactic they used. However, such an approach is sound only if females consistently use the same maternal care tactic, i.e., if the behavior is repeatable. Therefore, we performed a GLMM-based repeatability analysis[61] on the duration of maternal care for females with at least two observed maternal care periods. A multiplicative overdispersion model for binary data with a logit function was constructed, allowing the extraction of the within-individual variance and the components of residual variance necessary for the estimation of repeatability on the latent ($r_{logit}$) and original ($r_{original}$) scales. However, we only present the results on the original scale. The 95% confidence intervals were constructed using a bootstrapping procedure comprising 1000 iterations. Analyses were performed using the R package "rptR"[61]. To test whether differences in maternal care tactic could be due to maternal experience, we first tested whether age differed between females using a consistent maternal care tactic and females that alternated between tactics over the study period using a Student's *t*-test. Second, we verified whether the probability of using either one maternal care tactic (1.5-year tactic coded "0," 2.5-year tactic coded "1") was related to female reproductive status (primiparous or multiparous) using a generalized linear mixed effects model (R package "lme4"[62]), with a binomial distribution and female identity as a random factor. In six instances, female reproductive status could not

be determined and those data were removed from the above analysis. Then, females were classified into two groups according to the average duration of maternal care they provided; 13 females with average duration of maternal care ≥2 years were classified as using the 2.5-year tactic and 27 females with average duration of maternal care <2 years were classified as using the 1.5-year tactic. To increase sample size and because we found support for the repeatability of this trait, females with single reproductive events were also included in the demographic analyses and classified according to the tactic used in that event, for a total of 40 females using the 1.5-year tactic and 18 using the 2.5-year tactic.

Then, for each group of maternal care tactic, we estimated survival probability and recruitment rate (number of yearling daughters produced per year). We followed the recognized life cycle for this population with its respective age structure[34], except that our first age class was "yearling" to avoid assuming a 1:1 offspring sex-ratio in cubs-of-the-year that are not captured. The resulting population age-structure was thus: "yearling," "2 y.o.," "3 y.o.," "4–8 y.o.," and "9+ y.o." (Supplementary Fig. 1). We estimated survival for females within those five age classes. Fecundity represents the probability that a female survives and reproduces the next year (i.e., $Fecundity_t = Survival_{(t \to t+1)} \times Recruitment_{t+1}$). Therefore, we estimated recruitment for age classes 5–9 and 10+ y.o. to follow age classes for survival and because 5 y.o. is the youngest age at which females may start producing yearlings. Sample sizes used to estimate each tactic- and age class-specific demographic rates are presented in Supplementary Table 1.

Survival probability ("1" = survival, "0" = mortality) and recruitment (range: 0–3 yearling daughters per year) of females were compared between tactics and age classes using binomial and negative binomial (R packages "lme4" and "blme"[63]) generalized linear mixed effects models, respectively. We added an interaction between tactic and age class to compare within-age-class differences in survival, with year as a random effect. Female identity was added as a random effect in models of recruitment to account for multiple observations of the same females. Parameter significance tests and model simplifications were performed with likelihood ratio tests. Homoscedasticity was checked by plotting the residuals of the models. The resulting models were used to generate bootstrapped (10,000 iterations) tactic- and age-specific predictions of survival probability and recruitment to generate 95% confidence intervals using the R package "lme4"[62].

To estimate asymptotic population growth rate, $\lambda$, a proxy of fitness, for each maternal care tactic, we inserted these tactic- and age-specific model predictions for survival probability and recruitment into two different population models, one each tactic. We constructed two $9 \times 9$ female-based Leslie population matrix models ($A_{1.5}$ and $A_{2.5}$) comprising the five previously described age classes. For ages between 4 and 8, we added separated columns and rows even if single survival probabilities and fecundity were calculated for these age classes, resulting in $9 \times 9$ rather than $5 \times 5$ matrices, as follows:

$$A_{1.5} = \begin{bmatrix} 0 & 0 & 0 & F_{4-8,1.5} & F_{4-8,1.5} & F_{4-8,1.5} & F_{4-8,1.5} & F_{4-8,1.5} & F_{9+8,1.5} \\ S_{1,1.5} & 0 & 0 & 0 & 0 & 0 & 0 & 0 & 0 \\ 0 & S_{2,1.5} & 0 & 0 & 0 & 0 & 0 & 0 & 0 \\ 0 & 0 & S_{3,1.5} & 0 & 0 & 0 & 0 & 0 & 0 \\ 0 & 0 & 0 & S_{4-8,1.5} & 0 & 0 & 0 & 0 & 0 \\ 0 & 0 & 0 & 0 & S_{4-8,1.5} & 0 & 0 & 0 & 0 \\ 0 & 0 & 0 & 0 & 0 & S_{4-8,1.5} & 0 & 0 & 0 \\ 0 & 0 & 0 & 0 & 0 & 0 & S_{4-8,1.5} & 0 & 0 \\ 0 & 0 & 0 & 0 & 0 & 0 & 0 & S_{4-8,1.5} & S_{9+1.5} \end{bmatrix};$$

$$A_{2.5} = \begin{bmatrix} 0 & 0 & 0 & F_{4-8,2.5} & F_{4-8,2.5} & F_{4-8,2.5} & F_{4-8,2.5} & F_{4-8,2.5} & F_{9+8,2.5} \\ S_{1,2.5} & 0 & 0 & 0 & 0 & 0 & 0 & 0 & 0 \\ 0 & S_{2,2.5} & 0 & 0 & 0 & 0 & 0 & 0 & 0 \\ 0 & 0 & S_{3,2.5} & 0 & 0 & 0 & 0 & 0 & 0 \\ 0 & 0 & 0 & S_{4-8,2.5} & 0 & 0 & 0 & 0 & 0 \\ 0 & 0 & 0 & 0 & S_{4-8,2.5} & 0 & 0 & 0 & 0 \\ 0 & 0 & 0 & 0 & 0 & S_{4-8,2.5} & 0 & 0 & 0 \\ 0 & 0 & 0 & 0 & 0 & 0 & S_{4-8,2.5} & 0 & 0 \\ 0 & 0 & 0 & 0 & 0 & 0 & 0 & S_{4-8,2.5} & S_{9+2.5} \end{bmatrix}$$

This prevented us from having to attribute a fixed proportion of individuals transiting between the two age classes over time. The matrices were used to extract the dominant eigenvalue, i.e., the asymptotic population growth rate ($\lambda$) for each tactic. As a second measure of fitness, we also extracted the net reproductive rate ($R_0$), which corresponds to the number of females an individual is expected to produce over its lifetime. Apart from these fitness-related measures, we calculated the generation time ($T$) and stable age structure of each maternal care tactic to provide a demographic comparison of theoretical populations consisting of either one of the two maternal care tactics. All fitness and demographic metrics were calculated using the R package "popbio"[64]. For each tactic-based matrix model, we recalculated these metrics using bootstrapped model predictions for all survival probabilities and recruitment values to generate 95% confidence intervals.

**Differential effects of hunting between tactics**. To assess the influence of hunting pressure on the fitness of maternal care tactics, we first estimated an annual index of hunting pressure. This index was calculated by dividing the number of marked bears shot in a given year by the number of marked bears available for hunting that same year, i.e., the probability that a marked bear available for hunting was shot (Supplementary Fig. 2). Individuals within family groups are not available for hunting and thus were not considered in the index. Similarly, we excluded individuals that died before the onset of the hunting season.

Then, we modeled the effect of hunting pressure on each survival probability independently using Bayesian models with the R package "MCMCglmm"[65] with uninformative inverse-Wishart priors, 2,600,000 iterations, a thinning of 2500 and a burning of 100,000. In these models, survival ("1" = survival, "0" = death) was set as response variable and hunting pressure as the explanatory variable. Effect sizes (log-odds) of hunting on the survival probabilities of each maternal care tactic and each age-class are presented in Supplementary Table 2. We used these models to generate 1000 posterior predictions for each survival estimate over the range of hunting pressures observed (12 values of hunting pressure ranging between 0 and 0.33). These predictions were back-transformed on the original scale and introduced in the previously built tactic-specific matrix models. We repeated this procedure using the 95% posterior distribution of parameter estimates. This produced a posterior distribution of predicted $\lambda$ for each simulated hunting pressure and each maternal care tactic.

To tease apart the effect of hunting pressure from other potential density-related factors, we have conducted similar simulations using a population density index derived from county-level scat collection conducted in Sweden[66]. The weighted values of individual bear scat locations were summed in 10 km × 10 km cells to produce spatially explicit bear densities. These densities were then corrected for temporal variations using the Large Carnivore Observation Index[66]. Finally, annual density cells were summed over the study area and scaled (from 0 to 1) to obtain an index of annual density that reflects temporal changes in bear population density[43,44]. As for hunting pressure, effect sizes (log-odds) of density (Supplementary Table 3) on survival components were then used to generate posterior predictions of $\lambda$ dependent on population density index. However, the derived bear density index was only available for the period 1998–2015. Therefore, we also re-ran the simulations on hunting pressure, but this time only predicting $\lambda$ for the hunting pressures encountered during the period 1998–2015, when both indexes of hunting pressure and population density were available. We then generated model predictions for tactic- and age class-specific survival probabilities for 12 values of hunting pressure and population density observed in the population during the period 1998–2015 (Supplementary Table 3 and Supplementary Fig. 3).

**Data availability**. The data used in this study are fully available upon request from the corresponding authors.

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

## Acknowledgements

We thank R. Bischof, G. Blanchet, and D. Gravel for their advice and comments. We thank M. Festa-Bianchet and M. Leclerc for their comments on earlier versions of the manuscript. We are also grateful to S. Frank for his help with the index of bear population density. J.V.d.W. was financially supported by the Fonds de Recherche du Québec-Nature et Technologies (grant number: 184518) and by the Natural Sciences and Engineering Research Council of Canada (NSERC grant number: PGSD2–504356–2017). G.P. and F.P. were funded by NSERC discovery grant and by the Canada Research Chair in Evolutionary Demography and Conservation. This is scientific paper number 248 from the Scandinavian Brown Bear Research Project, which is funded by the Swedish Environmental Protection Agency, the Norwegian Directorate for Nature Management, the Austrian Science Fund, and the Swedish Association for Hunting and Wildlife Management. We acknowledge the support of the Center for Advanced Study in Oslo, Norway, that funded and hosted our research project "Climate effects on harvested large

mammal populations" during the academic year of 2015–2016 and funding from the Polish-Norwegian Research Program operated by the National Center for Research and Development under the Norwegian Financial Mechanism 2009–2014 in the frame of Project Contract No POL-NOR/198352/85/2013.

## Author contributions

J.V.d.W., F.P., A.Z. and J.E.S. have conceived the idea, J.V.d.W. and G.P. realized the analyses, J.V.d.W. wrote the first version of the manuscript and all authors have contributed to the subsequent versions of the manuscript. F.P. and A.Z. supervised the study. J.E.S. and A.Z. coordinated the Scandinavian Brown Bear Research Project.

## Additional information

**Competing interests:** The authors declare no competing interests.

