## [Peer Review File(PDF 535 kb) · Nature Communications]

Reviewers' comments:

Reviewer #1 (Remarks to the Author):

The authors derive demographic models for two reproductive strategies in brown bears based on empirical data. They show that because hunters avoid killing mothers with cubs, a longer weaning period offers a significant survival benefit for both mothers and their offspring. I find this manuscript very interesting and novel, although not novel in exactly the same way as the authors present it. The novelty is in combining empirical data with careful demographic analysis, in a context of harvest regime where the weaning period affects the chances of survival for both mothers and their cubs. It is not in studying "not size selective" harvest (which is a very broad category including unselective harvest, and dependencies on all sorts of characteristics other than size), nor in empirical analysis of harvest regime that provides some protection for weaning mothers (done by Tiilikainen et al. 2010).

General comments

To me, definition of "slow life history" is one that has longer generation time than some baseline. Longer inter-clutch intervals imply this, but given that the term is used prominently in the manuscript, it would be nice to actually show this. Given that you already have your Leslie matrix done, this is a trivial task.

I do not quite agree with the statement "potential selective effects of harvest that is not size selective have received much less attention in the literature". Term "not size selective" creates a strange contrast, and "much less" is rather subjective. The first sentence of your abstract is a general statement that applies to selective and unselective harvest alike, so the ability to make such statements already suggests that "not size selective" is not a good selling point. Some wildlife models like Thelen 1991 and Proaktor et al. 2007 allow for size-independent harvest. Among the more empirically oriented papers, Tiilikainen et al. 2010 compared two harvest regimes, the latter one being similar to what Swedish bears experience. In fisheries literature, there is quite some recent attention on selection on behavioral traits (reviewed in Heino et al. 2015). Earlier theoretical models often considered stage-specific harvesting (e.g., Law and Grey 1989, Ernande et al. 2004), even though more recent models seem almost always contain some element of size dependence.

You have chosen to use (density-independent) population growth ratio as your fitness measure. If the population is growing, as it is in your case, then offspring born early in life are more worth than offspring born late. If the population is stable, then it does not matter. To get rid of this timing effect, you could use expected reproductive success (R_0) as your fitness measure. While the difference is not great for weak growth, it would be interesting to see anyway. See Mylius and Diekmann (1995) for a discussion on "proper" fitness measures.

The result that hunting can *increase* population growth ratio for a given reproductive strategy is strange (Figure 4b, Extended Data Figure 4a). I can imagine that this could happen in a model involving density-dependent feedbacks, but not in a simple Leslie model.

This suggests either a mistake, or some interesting consequence of how stochasticity is incorporated in the projections. If the latter, this needs to be elaborated.

Could there be carry-over effects of short versus long weaning period? For example, better condition or earlier maturation? I would add at least mention this possibility in the discussion.

There is a need to clean up the terminology. Terms "mortality rate" and "survival rate" are ambiguous. Often these are used to refer to probabilities instead of rates. If, however, they actually are rates, then the time dimension is missing. Reading the methods suggests that your "rates" are probabilities, but it is difficult to be sure. The interpretation of numerical values is different for rates and probabilities, so a reader should know this. The solution is simple: refer probabilities as such; if you use rates, provide the dimension whenever numeric values are reported. I would also refer population growth rate as population growth ratio, reproductive rate as fertility - neither is a rate, in sense of having a time dimension.

A related technical issue is that for probabilities, it is not very meaningful to say that "p2 is 5% larger than p1". An easy way of seeing this is that while in some sense $p_2=0.3$ is 50% larger than $p_1=0.2$, there is no p_2 that in that same sense is 50% larger than, say, $p_1=0.8$. The proper way of doing this is to use odds and odds ratios. Not everybody is accustomed to odds and odds ratios, but that is the only proper way of comparing probabilities. Notice that this is what logistic regressions do - the linear predictor is in $\log(\text{odds})$.

Specific comments

L42 This statement is true even when resources are not limited

L57 I would say that Tiilikainen et al. 2010 did so, although less elegantly than you

L83 it would be more informative to report the odds ratio than the Z

L83 "total mortality rate" is ambiguous - do you mean probability, or an (instantaneous) rate? If the first, make that explicit. If the second, the unit is missing (yr^{-1}). The same for all "mortality rates".

L84 report SE for 0.14

L88 meaningful precision: "71%" (this comparison is only meaningful for rates; if the values are probabilities, then you need to report the odds ratio)

L91 I would consistently report either SE's (as above) or CI's. CI's are nicer in that they show the possible asymmetry

L123 The values from Table 1 suggest that the correct fertilities are 0.260 and 0.397, but maybe I'm misinterpreting the results. If so, the table is not explained well enough.

L223 You should say somewhere that (you assume that) females become pregnant again during the season when weaning occurs, without a break

L265,283 These are not "age classes" but "age class groups"

L267 "fecundity" should only have subscript "t" and "reproduction" should only have subscript "t+1", right?

L285 "this age class" -> "these age classes"

Table 1 What does recruitment "-0.923" mean? These estimates should be back-transformed to readily interpretable values.

Figure 2b The y-axis labelling and title do not match - surely annual mortality is $>0.2\%$?

Figure 4a Under which level of hunting, if any?

Extended Data Table 1 What does "entire population" mean here? Including males?

Extended Data Table 2,3 Are the reported values log-odds? Transforming them to odds would be more informative (I know nobody who can easily exponentiate in head!).

Extended Data Figure 2 This is understandable but not technically correct as it implies than an individual could stay in the "4-8y.o." group forever. The "4-8y.o." node cannot have arrow back to itself. Instead, the node should be repeated five times (perhaps as overlapping nodes).

Extended Data Figure 4a It would be helpful to point out how is this different from Figure 4b (it is difficult to spot the difference in the year range).

Ernande, B., Dieckmann, U., and Heino, M. 2004. Adaptive changes in harvested populations: plasticity and evolution of age and size at maturation. *Proceedings of the Royal Society of London B: Biological Sciences*, 271: 415–423.

Heino, M., Diaz Pauli, B., and Dieckmann, U. 2015. Fisheries-induced evolution. *Annual Review of Ecology, Evolution, and Systematics*, 46: 461–480.

Law, R., and Grey, D. R. 1989. Evolution of yields from populations with age-specific cropping. *Evolutionary Ecology*, 3: 343–359.

Mylius, S. D., and Diekmann, O. 1995. On evolutionarily stable life histories, optimization and the need to be specific about density dependence. *Oikos*, 74: 218–224.

Proaktor, G., Coulson, T., and Milner-Gulland, E. J. 2007. Evolutionary responses to harvesting in ungulates. *Journal of Animal Ecology*, 76: 669–678.

Thelen, T. H. 1991. Effects of harvest on antlers of simulated populations of elk. *Journal of Wildlife Management*, 55: 243–249.

Tiilikainen, R., Nygrén, T., Pusenius, J., and Ruusila, V. 2010. Variation in growth pattern of male moose *Alces alces* after two contrasted periods of hunting. *Annales Zoologici Fennici*, 47: 159–172.

Reviewer #2 (Remarks to the Author):

This manuscript approaches the 'harvest selection' body of work from an under-examined perspective and offers a significant empirical contribution. Whereas most work has focused on how size-selective harvesting by 'human predators' (ie hunters) can (often rapidly) reduce size-at-age or speed up reproductive schedules, this paper shows how the opposite effect (slowed reproductive schedules) is possible, owing to certain reproductive classes (females with young, and the accompanying young) being safeguarded from harvests. Although I am ill-equipped to evaluate the statistical analysis (owing to lack of this expertise), no alarm bells go off. The authors build (on sound theoretical footing) from the

ideas of their own and others and confront the hypothesis with a long-term (20+ year) detailed data set. I believe this paper will generate significant interest and citations.

Although I do believe a wide readership would be interested, the authors could consider increasing the generalizability of this manuscript. Some might argue that grizzly bears represent just a handful of wildlife systems in which this process could operate. Papers on moose, chamois and grizzlies are cited, but I suspect that hunters forgoing females with young (in systems in which there is variation in length of maternal care) might apply to many more terrestrial wildlife for regulatory (or cultural) reasons. Also, the authors hypothesize that infanticide might be one factor (unexplored but likely important) contributing to the effect (via post-care offspring of 1.5 year' mothers being killed at higher rates than post-care offspring of the 2.5 year tactic. Infanticide, however, is relatively rare. Perhaps the authors can make the general argument that offspring released earlier likely, all things equal, suffer higher mortality rates than those released from care later.

Relatedly, the authors could consider making the point that the selective landscape imposed by humans, as often dominant agents of mortality, can be as varied as the phenotypic targets, which is not only restricted to ornaments and body size [common dimensions explored]. Although some of this thinking occurs in the early part of the manuscript, the authors return to this theme with a more developed argument.

I am surprised that the alternative hypothesis (decrease in resources, line 75) to explain the increased frequency of the 2.5 yr tactic is so rapidly and thinly dismissed. I feel comfortable with it, especially when your growth rate modeling results show increases among the 'sub-population' comprised of this phenotype with increased hunting pressure. Similarly, you show no effect when growth rates are modeled against population density. But surely someone on this bear project or someone else that has done work in the area has data about the (forage) productivity over the time period? Even a citation would be helpful.

Life history traits can have high heritability. Are there enough data on marked individuals of known descent (and repeatable expression of tactic) to calculate the heritability? An estimate could enrich your (conceptual) model, discussion and argument.

37.5% of females varied in tactic across litters. Was there any directionality with age?

Another curiosity question: do litter sizes vary between tactics? This information is folded up (and not shown) in the reproductive output data. This also directly links to the content of the Zedrosser et al 2011 paper, that showed that a history of longer persecution (ie European bears vs new world bears) had higher litter sizes when accounting for female body mass.

There is a focus ('crucial question', line 125) on which phenotype yields the highest fitness. Pls consider modifying that discussion around the likelihood that the two phenotypes likely will (and have a history of) varying in their relative frequencies as the environment (resources, mortality regime) changes. Its actually an interesting result that (under the last 20 years) the two have similar fitness. This might be why the phenotypes have been

maintained. Its an argument about the maintenance of biological (life history) diversity. You get there in lines 145 with discussion on 'promoting...alternative maternal care tactics', but it comes late as is overshadowed by the focus on 'optimality' (eg line 295).

Line 83 and others – pls make clear that these are annual finite rates of mortality

I would consider formally combining the Results and Discussion. You include lots of discussion matter before the 1-paragraph 'Discussion' section at the end.

Chris Darimont

Reviewers' comments:

Reviewer #1 (Remarks to the Author):

The authors derive demographic models for two reproductive strategies in brown bears based on empirical data. They show that because hunters avoid killing mothers with cubs, a longer weaning period offers a significant survival benefit for both mothers and their offspring. I find this manuscript very interesting and novel, although not novel in exactly the same way as the authors present it. The novelty is in combining empirical data with careful demographic analysis, in a context of harvest regime where the weaning period affects the chances of survival for both mothers and their cubs. It is not in studying "not size selective" harvest (which is a very broad category including unselective harvest, and dependencies on all sorts of characteristics other than size), nor in empirical analysis of harvest regime that provides some protection for weaning mothers (done by Tiilikainen et al. 2010).

***Response:** We agree with the reviewer and we now emphasize that the novelty of our work is in showing the potential of a harvest regime protecting females based on their reproductive status to affect female life-history traits and population processes. We have modified the text in the introduction (L63-65, L80-87) and conclusion (L277-279) following the reviewer's suggestion. We now also cite the work of Tiilikainen et al. 2010 (L43, L50, L52 and L62).*

General comments

To me, definition of "slow life history" is one that has longer generation time than some baseline. Longer inter-clutch intervals imply this, but given that the term is used prominently in the manuscript, it would be nice to actually show this. Given that you already have your Leslie matrix done, this is a trivial task.

***Response:** Following the reviewer's recommendation, we now provide an estimate of generation time for the two maternal care tactics and show that hunting regulation should promote slower life histories through a lengthening in generation time (L156-L164).*

I do not quite agree with the statement "potential selective effects of harvest that is not size selective have received much less attention in the literature". Term "not size selective" creates a strange contrast, and "much less" is rather subjective. The first sentence of your abstract is a general statement that applies to selective and unselective harvest alike, so the ability to make such statements already suggests that "not size selective" is not a good selling point. Some wildlife models like Thelen 1991 and Proaktor et al. 2007 allow for size-independent harvest. Among the more empirically oriented papers, Tiilikainen et al. 2010 compared two harvest regimes, the latter one being similar to what Swedish bears experience. In fisheries literature, there is quite some recent attention on selection on behavioral traits (reviewed in Heino et al. 2015). Earlier theoretical models often

considered stage-specific harvesting (e.g., Law and Grey 1989, Ernande et al. 2004), even though more recent models seem almost always contain some element of size dependence.

Response: *We agree with the reviewer that recent work has shown more focus on the selective potential of non size-selective harvest. We have modified the first paragraph of the introduction to include all the citations on theoretical and empirical work provided by the reviewer (L41-43). We also changed the term 'not size selective' for 'size-independent' (L41) and avoided the use of the term "much" (L40).*

You have chosen to use (density-independent) population growth ratio as your fitness measure. If the population is growing, as it is in your case, then offspring born early in life are more worth than offspring born late. If the population is stable, then it does not matter. To get rid of this timing effect, you could use expected reproductive success (R_0) as your fitness measure. While the difference is not great for weak growth, it would be interesting to see anyway. See Mylius and Diekmann (1995) for a discussion on "proper" fitness measures.

Response: *Following the reviewer's suggestion, we now use population growth rate, λ , as well as expected reproductive success, R_0 , (referred to as "net reproductive rate" in the manuscript following the terminology used in Caswell (2001)) as our measures of fitness to contrast the two maternal care tactics (see Fig. 4b in the revised version of the manuscript and L72-76, L150-152). We also included the reference to the work of Mylius and Diekmann (1995) at L72 and L75 to justify our use of the two fitness proxy.*

The result that hunting can *increase* population growth ratio for a given reproductive strategy is strange (Figure 4b, Extended Data Figure 4a). I can imagine that this could happen in a model involving density-dependent feedbacks, but not in a simple Leslie model. This suggests either a mistake, or some interesting consequence of how stochasticity is incorporated in the projections. If the latter, this needs to be elaborated.

Response:

We agree with the reviewer that the apparent increase in population growth rate with hunting pressure is unexpected. However, this does not arise due to a mistake, but as a consequence of how uncertainty was incorporated into the projections.

To test how hunting pressure affects population growth rate, we have allowed survival probabilities for each age class (5 in total) to vary according to hunting pressure in our Leslie matrix models. Because the effect of hunting pressure can differ between age-classes and between maternal care tactics (adult females using the 2.5-year tactic should be less affected by hunting pressure compared to the 1.5-year tactic), we have constructed a different Bayesian model for each age-class of each maternal care tactic. Because we had 5 age classes, we have constructed 5 different Bayesian models for the 1.5-year tactic and 5 models for the 2.5-year tactic. However, because all yearlings from the 2.5-year maternal care tactic survived, we did not construct a model for the survival probabilities of yearlings from the 2.5-year tactic and have kept it constant as 1.0 in all simulations, which resulted in a total of 9 Bayesian models. In our models,

survival (1=survival, 0=death) was set as the response variable and hunting pressure as an explanatory variable. Survival is a binomial process and models were constructed using a logit function. We then used this posterior distribution of the slope and intercept to predict tactic-specific survival probabilities for 12 hunting pressures, ranging from 0 to 0.33, which is the range of hunting pressures observed in the population over the course of the study. This resulted in the prediction of 1,000 normally distributed survival probabilities (on the logit scale) for each survival rate for each maternal care tactic and each hunting pressure simulated. However, to produce comprehensive estimations of population growth rate, those estimates were back-transformed to their original scale prior to being incorporated into our Leslie matrices. Those back-transformed predictions were then incorporated into Leslie matrix models to extract a distribution of population growth rates (λ) for each tactic at each hunting pressure.

The skewness of the distribution of λ (for any given hunting pressure, see Figure 4B in the previous version of the manuscript) arises from this back-transformation. When hunting pressure is low, predictions for survival rates are high, which results in a skewed distribution of survival probability when back-transforming on the original scale (see Figure A for an example).

Figure A. Effect of back-transforming model predictions for survival on the original scale. The example shows the survival probabilities of females within the age-class 9+ y.o for the 2.5-year tactic at a hunting pressure of 0.

Adding the effect of hunting pressure causes the predictions for survival to decline on the logit scale, but it also reduces the skewness in the distribution of back-transformed survival probabilities (see Figure B for an example).

Figure B. Effect of back-transforming model predictions for survival on the original scale. The example shows the survival probabilities of females within the age-class 9+ y.o for the 2.5-year tactic at a hunting pressure of 0.09.

This reduces the probability of obtaining very small values as hunting pressure increases. As a result, combinations of survival probabilities included in our Leslie matrices contain fewer and fewer extremely small values as hunting pressures increases, which contributes to reducing the skewness in λ estimates and causes the apparent increase in population growth rate in the first half of the curve. However, the overall distribution continues to shift towards smaller values as hunting pressure increases (see Figure C for an example), which explains the decline in population growth rate for the 2.5-year tactic in the second half of the curve in original Figure 4B.

Figure C. Effect of back-transforming model predictions for survival on the original scale. The example shows the survival probabilities of females within the age-class 9+ y.o for the 2.5-year tactic at a hunting pressure of 0.33.

In the original Figure 4b, we have forced a line for each tactic that passes through the means of the posterior distribution of lambda for each hunting pressure. These lines were added for ease of interpretation, however, because the skewness of lambda

changes as a function of the combination of multiple back-transformed survival probabilities, drawing the mean of the posterior distribution of lambda on the plot may be misleading. In fact, if we only consider the mean predictions for each survival probability (rather than the entire distribution) in our simulations, the apparent increase in lambda as hunting pressure increases vanishes (see Figure D).

Figure D: Changes in population growth rate (lambda, λ) as a function of hunting pressure for the two maternal care tactics (“1.5-year tactic”; open circles, “2.5-year tactic”; black circles).

However, because the mean of logs does not equal the log of means, we cannot present both measures on the figure. Thus, and after consulting with a specialist in mathematical modeling (Prof. Dominique Gravel, Université de Sherbrooke), we have decided to report solely the hunting pressure-specific distribution of lambda for each tactic to provide the reader with the full range of possibilities rather than a unique value. Also, after consultation with a biostatistician (Dr. Guillaume Blanchet, Université de Sherbrooke), we have chosen to use the 95% credible interval (rather than 100%) of the posterior distribution of parameters to avoid unrealistic predictions and reduce the skewness in our projections. All these improvements were applied to Fig. 5 and Supplementary Fig. 3 in the new version of the manuscript to facilitate interpretation.

An increase in lambda with hunting pressure also appears in the original Extended data Figure 4a, where we intended to compare the effect of hunting pressure with the effect of population density on the population growth rate of both tactics. However, the index of population density was unavailable before 1998 (and our analyses are based on monitoring data from 1993-2015). Therefore, we were limited to the period 1998-2015 to test the effect of population density on population growth rate. To temporally match the data on population density, we provided (in the original version of the manuscript) a similar analysis on the effect of hunting pressure on population growth rate of both tactics using only survival information from 1998-2015. However, within the 2.5-year tactic, only one female aged 4-8 y.o. died during the monitoring period

(out of 65 from 1993-2015 and out of 60 from 1998-2015). This mortality occurred under a hunting pressure of 0.18, which is an intermediate value for the period 1993-2015, but a low value for the period 1998-2015. By restricting the time period to 1998-2015, this resulted in an overestimation of the effect of hunting pressure on the survival probability of females aged 4-8 y.o. ($\beta=25.118$, 95%CI = [-20.582, 72.927]; Extended data Table 3 in the original version of the manuscript), hence the apparent increase in population growth rate with hunting pressure (Figure E). By removing the effect of hunting pressure on the survival probability of females aged 4-8 y.o. within the 2.5-year tactic in our models, we do not observe an increase in population growth rate with hunting pressure (Figure F).

Figure E: Effect of hunting pressure on population growth rate for the two maternal care tactics when considering the effect of hunting on the survival rate of females aged 4-8 y.o. within the 2.5-year tactic.

Figure F: Effect of hunting pressure on population growth rate for the two maternal care tactics when removing the effect of hunting on the survival rate of females aged 4-8 y.o. within the 2.5-year tactic.

Therefore, to avoid any artificial overestimation of the effect of hunting pressure on the survival probability of females aged 4-8 y.o. within the 2.5-year tactic, we have used the model coefficients for the effect of hunting pressure on survival for the entire period, i.e. 1993-2015, but predicted survival probabilities only for the hunting pressures observed during the period 1998-2015 (Supplementary Fig. 3 in the revised version of the manuscript).

Could there be carry-over effects of short versus long weaning period? For example, better condition or earlier maturation? I would add least mention this possibility in the discussion.

Response: The analyses presented in the manuscript allow for the interpretation of the effect of maternal care tactic on the survival probabilities of yearling, 2y.o., and 3y.o. females only. The survival probability for age classes 2 y.o. and 3 y.o. was similar between tactics, suggesting the absence of carry-over effects of the duration of maternal care on sub-adult female survival. Following the recommendation, we now discuss the possibility of carry-over effects of prolonged maternal care in L220-226.

There is a need to clean up the terminology. Terms "mortality rate" and "survival rate" are ambiguous. Often these are used to refer to probabilities instead of rates. If, however, they actually are rates, then the time dimension is missing. Reading the methods suggests that your "rates" are probabilities, but it is difficult to be sure. The interpretation of numerical values is different for rates and probabilities, so a reader should know this. The solution is

simple: refer probabilities as such; if you use rates, provide the dimension whenever numeric values are reported. I would also refer population growth rate as population growth ratio, reproductive rate as fertility - neither is a rate, in sense of having a time dimension.

Response: *Following the reviewer's suggestion, we provide the dimension when reporting rates (the dimension here is 1 year for every rate calculated) and we refer to probabilities when referring to survival estimates from logistic models. Regarding population growth rate, we estimated the dominant eigenvalue of our Leslie population matrices, λ , which is referred to as the population growth rate in population ecology (Examples: Saether & Bakke 2000; Caswell 2001; Sibly & Hone 2002; Bieber & Ruf 2005). It represents the per capita rate of population increase (Sibly & Hone 2002), i.e. the number by which each female within the population is replaced at each time step (here, a year) at equilibrium. Therefore, because λ , as used here, has a time dimension and for consistency purposes with published literature on population ecology, we prefer to keep the term "population growth rate" when directly referring to λ and we now define it at first mention (L73-74). However, we privilege the use of the term "population growth" over "population growth rate" elsewhere in the text. Relatedly, and more generally, the terminology we used in the manuscript follows that of Caswell (2001), with all components of the Leslie matrix being referred to as "vital rates". To estimate fecundity, we multiplied the survival probability of females from age i to age $i+1$ by the recruitment rate of females of age $i + 1$. We now avoid the use of reproductive rate and only refer to recruitment rate, which is now defined at L122 as the number of yearling daughters produced per year per female. Using this definition it can be considered as a rate with a time dimension (1 year).*

A related technical issue is that for probabilities, it is not very meaningful to say that "p2 is 5% larger than p1". An easy way of seeing this is that while in some sense $p_2=0.3$ is 50% larger than $p_1=0.2$, there is no p_2 that in that same sense is 50% larger than, say, $p_1=0.8$. The proper way of doing this is to use odds and odds ratios. Not everybody is accustomed to odds and odds ratios, but that is the only proper way of comparing probabilities. Notice that this is what logistic regressions do - the linear predictor is in $\log(\text{odds})$.

Response: *We would like to thank the reviewer for this insightful comment. Following the reviewer's recommendation, we now use odds and odd ratios in our comparisons of survival probabilities (L96-98, L104-106, L126-127).*

Specific comments

L42 This statement is true even when resources are not limited

Response: *The sentence has been modified in consideration of the reviewer's comment (L45-46).*

L57 I would say that Tiilikainen et al. 2010 did so, although less elegantly than you

Response: *We have modified the statement and cited the work of Tiilikainen et al. (2010) at L59-62.*

L83 it would be more informative to report the odds ratio than the Z

Response: *Recommendation followed (L96-98).*

L83 "total mortality rate" is ambiguous - do you mean probability, or an (instantaneous) rate? If the first, make that explicit. If the second, the unit is missing (yr⁻¹). The same for all "mortality rates".

Response: *Following recommendation, the sentence has been modified to specify the unit (L98-104).*

L84 report SE for 0.14

Response: *Considering the below comment from the reviewer, we now provide the 95% CI (L99-100).*

L88 meaningful precision: "71%" (this comparison is only meaningful for rates; if the values are probabilities, then you need to report the odds ratio)

Response: *Recommendation followed (L104-106)*

L91 I would consistently report either SE's (as above) or CI's. CI's are nicer in that they show the possible asymmetry

Response: *Recommendation followed. We have reported only confidence intervals at L96-104 and in Table 1.*

L123 The values from Table 1 suggest that the correct fertilities are 0.260 and 0.397, but maybe I'm misinterpreting the results. If so, the table is not explained well enough.

Response: *Table 1 presents the results from models explaining variation in survival and recruitment between maternal care tactics. As such, it aims at presenting results from statistical testing. Based on these models, we calculated predictions of survival for each age class and tactic and predictions of recruitment for each tactic. We used a bootstrap approach, which generated a total of 10,000 predictions for each parameter. We have reported the mean and 95% CI of the bootstrap distribution for each parameter in Table 2, which may slightly differ from the direct back-transformation of the model coefficients in Table 1. The correct recruitment rates for tactic 2.5 and 1.5 should have been 0.251 and 0.384 (the mean bootstrap recruitment estimates shown in Table 2). We have corrected these numbers in the text (L141-144) and clarified the titles for both Table 1 and Table 2 in the revised version of the manuscript.*

L223 You should say somewhere that (you assume that) females become pregnant again during the season when weaning occurs, without a break

Response: *In our study population, 95% of females give birth to a new litter after weaning or losing their litter. We have added this information (unpublished data) in L309-312 as requested.*

L265,283 These are not "age classes" but "age class groups"

Response: *Throughout the manuscript, we rely on the terminology of stage-classified matrix population models (Caswell 2001), where age is a special case of stage-classified models. Therefore, stage is defined by age composition. Following this terminology, individuals are classified within age classes according to the species' life cycle. An age class can include several years (Caswell 2001; Otto & Day 2007; Pelletier et al. 2011; Gervasi et al. 2012; Newman et al. 2014). We thus believe that our usage of the term "age classes" is correct in this context.*

L267 "fecundity" should only have subscript "t" and "reproduction" should only have subscript "t+1", right?

Response: *Yes. We made the appropriate changes (L367). Also, following another reviewer comment, we now refer to "recruitment" rather than "reproduction".*

L285 "this age class"->"these age classes"

Response: *Corrected (L387).*

Table 1 What does recruitment "-0.923" mean? These estimates should be back-transformed to readily interpretable values.

Response: *This table aims at showing the statistical differences in demographic parameters between the two maternal care tactics. The coefficients presented in Table 1 are thus reported on the transformed scale to show parameter significance. For recruitment, we used a negative binomial model. Back-transforming -0.923 on its original scale requires taking its anti-log, i.e. exponentiating. This results in a recruitment rate of 0.397 yearling daughters per year per adult female. However, the final models presented in Table 1 were used to compute model-based predictions of survival and recruitment. Those predictions were back-transformed on their original scale (readily interpretable values) and are already presented in Table 2. The titles of Table 1 and Table 2 were adjusted to provide more details and avoid confusion.*

Figure 2b The y-axis labelling and title do not match - surely annual mortality is >0.2%?

Response: *Figure 2b has been changed so that the y-axis labelling and the title now match.*

Figure 4a Under which level of hunting, if any?

Response: *The bootstrapped distributions represented in Figure 4a were obtained using all empirical data covering the entire study period 1993-2015. Therefore, they cover a range of hunting pressures, rather than a specific level. To avoid confusion with the panel **b**, where the distributions were calculated over various levels of hunting pressure, we have separated the panels **a** and **b**. The new Fig. 4 now aims at showing the general differences between the two maternal care tactics in terms of asymptotic population growth rate, λ , and population stage age structure. Also, following the reviewer's suggestion from a previous comment, we now include a comparison of net reproductive rate, R_0 , and generation time between the two maternal care tactics.*

Extended Data Table 1 What does "entire population" mean here? Including males?

Response: *We apologize for the confusion. "Entire population" refers to all females in the population, regardless of their maternal care tactic. It also includes females that were excluded from the tactic-based models, because we were unable to classify them within one tactic or the other for several reasons, for example, when we were not able to accurately determine weaning time. The column name was changed to "All females" and the table legend was modified and now includes an explanation of what is meant by "All females" in the Supplementary Table 1 in the revised version of the manuscript.*

Extended Data Table 2,3 Are the reported values log-odds? Transforming them to odds would be more informative (I know nobody who can easily exponentiate in head!).

Response: *Estimates of the slopes in the table are given on the logit scale (they are thus log-odds), which is the scale on which survival probability for each age-class and maternal care tactic as a function of hunting pressure was modeled. The aim of this table was to show that hunting pressure had a different effect on the survival probabilities of females based on their maternal care tactic. Thus, the aim was not to provide effect size. We have chosen to keep the reported values on the logit scale, because it answers our purposes and because not everyone is accustomed with models results tables showing odds and odds ratio and we wish to keep our manuscript accessible to a vast readership.*

Extended Data Figure 2 This is understandable but not technically correct as it implies than an individual could stay in the "4-8y.o." group forever. The "4-8y.o." node cannot have arrow back to itself. Instead, the node should be repeated five times (perhaps as overlapping nodes).

Response: *In the presentation of the stage- (or age-) structured life cycle graph, we referred to the work of Caswell (2001) and followed steps 1-5 (page 56-57) in which it is stated that: "If an individual in stage i at time t can contribute to stage i at time $t+1$ (e.g. by remaining in the same stage from one time to the next), put an arc from N_i to itself; such an arc is called a self-loop". Also, a similar life cycle graph from the same*

population has also been published elsewhere (Gosselin et al. 2015). Therefore, we prefer to keep the life cycle graph (now Supplementary Figure 1) as is in the current manuscript.

Extended Data Figure 4a It would be helpful to point out how is this different from Figure 4b (it is difficult to spot the difference in the year range).

Response: *Recommendation followed. The year range is now indicated in the figure title and we have adjusted the figure legend to point out how Extended Data Figure 4a (now Supplementary Fig. 3a) differs from Figure 4b (now Fig. 5).*

Ernande, B., Dieckmann, U., and Heino, M. 2004. Adaptive changes in harvested populations: plasticity and evolution of age and size at maturation. Proceedings of the Royal Society of London B: Biological Sciences, 271: 415–423.

Heino, M., Diaz Pauli, B., and Dieckmann, U. 2015. Fisheries-induced evolution. Annual Review of Ecology, Evolution, and Systematics, 46: 461–480.

Law, R., and Grey, D. R. 1989. Evolution of yields from populations with age-specific cropping. Evolutionary Ecology, 3: 343–359.

Mylius, S. D., and Dieckmann, O. 1995. On evolutionarily stable life histories, optimization and the need to be specific about density dependence. Oikos, 74: 218–224.

Proaktor, G., Coulson, T., and Milner-Gulland, E. J. 2007. Evolutionary responses to harvesting in ungulates. Journal of Animal Ecology, 76: 669–678

Thelen, T. H. 1991. Effects of harvest on antlers of simulated populations of elk. Journal of Wildlife Management, 55: 243–249.

*Tiilikainen, R., Nygrén, T., Pusenius, J., and Ruusila, V. 2010. Variation in growth pattern of male moose *Alces alces* after two contrasted periods of hunting. Annales Zoologici Fennici, 47: 159–172.*

Response: *We thank the reviewer for the references and we have included all of them in the new version of the manuscript.*

Reviewer #2 (Remarks to the Author):

This manuscript approaches the ‘harvest selection’ body of work from an under-examined perspective and offers a significant empirical contribution. Whereas most work has focused on how size-selective harvesting by ‘human predators’ (ie hunters) can (often rapidly) reduce size-at-age or speed up reproductive schedules, this paper shows how the opposite effect (slowed reproductive schedules) is possible, owing to certain reproductive classes (females with young, and the accompanying young) being safeguarded from harvests. Although I am ill-equipped to evaluate the statistical analysis (owing to lack of this expertise), no alarm bells go off. The authors build (on sound theoretical footing) from the ideas of their own and others and confront the hypothesis with a long-term (20+ year) detailed data set. I believe this paper will generate significant interest and citations.

Response: *We thank the reviewer for this positive comment.*

Although I do believe a wide readership would be interested, the authors could consider increasing the generalizability of this manuscript. Some might argue that grizzly bears represent just a handful of wildlife systems in which this process could operate. Papers on moose, chamois and grizzlies are cited, but I suspect that hunters forgoing females with young (in systems in which there is variation in length of maternal care) might apply to many more terrestrial wildlife for regulatory (or cultural) reasons.

Response: *There is actually a lack of literature on harvest systems where there is both avoidance of killing of females with dependent young and variation in the duration of maternal care. However, we agree with the reviewer that our findings should apply to many harvest systems, as avoidance of females with dependent young is widespread among harvest systems and there are several game species where the duration of maternal care is variable. Therefore, following the reviewer’s suggestion, we have increased the generalizability of our manuscript at the end of the discussion section (L277-285), by providing more examples of hunting systems where it is prohibited (or hunters may avoid) to kill females with young and examples of species where the duration of maternal care varies. We also provide a perspective on future work by saying that the survival advantage of females with young could also select for other life-history traits, such as earlier age at reproduction.*

Also, the authors hypothesize that infanticide might be one factor (unexplored but likely important) contributing to the effect (via post-care offspring of 1.5 year’ mothers being killed at higher rates than post-care offspring of the 2.5 year tactic. Infanticide, however, is relatively rare. Perhaps the authors can make the general argument that offspring released earlier likely, all things equal, suffer higher mortality rates than those released from care later.

Response: *Following the reviewer’s suggestion, we changed the text by providing the more general argument that longer periods of maternal care can increase survival prospects of offspring and provide a citation on leopards (L193-199). While we agree that importance of infanticide as mortality factor can vary between brown bear populations, Sexually Selected Infanticide (SSI) is the most important cause of mortality*

for dependent young and can influence population growth rate in our study system (Swenson et al. 2001b; Gosselin et al. 2015). However, while the killing of dependent yearlings can happen (Swenson, Dahle & Sandegren 2001a), most intra-specific killings of yearlings occur once they are independent, i.e. after family break-up and the mating season in the spring, which suggests that SSI cannot explain those yearling mortalities. Mortality rate of independent yearling females due to intra-specific killing is about 16% in our study area, which suggests it is an important factor worth considering.

Relatedly, the authors could consider making the point that the selective landscape imposed by humans, as often dominant agents of mortality, can be as varied as the phenotypic targets, which is not only restricted to ornaments and body size [common dimensions explored]. Although some of this thinking occurs in the early part of the manuscript, the authors return to this theme with a more developed argument.

Response: *Following the reviewer's suggestion, we added a sentence in the beginning of the discussion making the point that the selective landscape imposed by humans, as dominant agents of mortality, can be as varied as the phenotypic targets, and we included a citation to support this claim (L177-178). However, we are unsure as to how to interpret the remaining of the reviewer's suggestion.*

I am surprised that the alternative hypothesis (decrease in resources, line 75) to explain the increased frequency of the 2.5 yr tactic is so rapidly and thinly dismissed. I feel comfortable with it, especially when your growth rate modeling results show increases among the 'sub-population' comprised of this phenotype with increased hunting pressure. Similarly, you show no effect when growth rates are modeled against population density. But surely someone on this bear project or someone else that has done work in the area has data about the (forage) productivity over the time period? Even a citation would be helpful.

Response: *Unfortunately, we do not have data on forage productivity for the period 1993-2015, hindering proper assessment of the effect of forage productivity on the duration of maternal care. However, following the reviewer's suggestion, we expand our explanation of why we do not expect that changes in resources abundance in the study population could have driven the observed increase in the frequency of the 2.5-year maternal care tactic (L227-236). We provide a reference from an article from members of the Scandinavian Brown Bear Project showing how bilberry abundance index has changed over the last ten years in the study area (Hertel et al. 2017). While there were important inter-annual variations in the estimated index of bilberry abundance and these variations were associated with variation in female reproductive success, there was no temporal decline in the abundance of bilberry over the last ten years. It has also been shown that Scandinavian brown bears can switch to alternative food items (i.e. crowberry) in years of low berry production, which makes them less vulnerable to food shortage (Stenset et al. 2016).*

Life history traits can have high heritability. Are there enough data on marked individuals of known descent (and repeatable expression of tactic) to calculate the heritability? An estimate could enrich your (conceptual) model, discussion and argument.

Response: We agree with the reviewer's comment, however our sample size is not large enough yet to provide this information. Heritability of the duration of maternal care could be estimated using quantitative genetics and animal models. We are now working on the building of the pedigree of the south-central Sweden brown bear population. Estimation of heritability for several morphological, behavioral and life-history traits are part of an ongoing investigation and we wish not to overlap with these projects. Preliminary results, however, suggest that we may not have enough statistical power to detect heritability values of the range expected for life-history traits (Figure F). This problem is even more pronounced in traits expressed only in females. The duration of maternal care is a trait that can only be expressed late in life, i.e. once a female has successfully produced and weaned a litter. In the Scandinavian brown bear, females start to successfully reproduce at an average age of 5.4 years-old (Swenson et al. 2001b) and if we consider an 1.5-year duration of maternal care, the trait of interest can be measured at the age of 7 years at the earliest. Considering that average age of mortality for female Scandinavian brown bears is 4.8 years (Zedrosser et al. 2013), we are highly limited in our sample size. At the current time, we only have 37 mother-daughter pairs for which we were able to measure this trait, which may be insufficient to statistically detect heritability for this trait. In addition, the occurrence of the 2.5-year tactic is relatively recent (since 1993, but more frequent only since 2005), which could further limit our ability to statistically detect heritability for this trait at this time.

Figure F: Statistical power to detect heritability of morphological (head circumference; grey), behavioral (bear localisations; black) and life-history traits (litter size: red, age at first reproduction: white) using the Scandinavian brown bear pedigree. Figure provided by Audrey Bourret, Université de Sherbrooke.

37.5% of females varied in tactic across litters. Was there any directionality with age?

Response: *Females that we considered to be consistent in their maternal care tactic were not older on average (mean = 10.9, 95% CI = [9.6, 12.1]) than more flexible females (mean = 12.1, 95% CI = [10.3, 13.8]; $t_{38} = -1.12$, $P = 0.27$), and primiparous females had a similar probability of using the tactic 1.5 compared to multiparous females (odds-ratio = 0.46, 95% CI = [0.05, 4.67], $n = 123$), suggesting that maternal experience may not be the main factor explaining differences in maternal care tactics. We have included these additional analyses in the methods section (L345-353) and the results in the main text (L113- 119).*

Another curiosity question: do litter sizes vary between tactics? This information is folded up (and not shown) in the reproductive output data. This also directly links to the content of the Zedrosser et al 2011 paper, that showed that a history of longer persecution (ie European bears vs new world bears) had higher litter sizes when accounting for female body mass.

Response: *If we focus only on successful reproductions (i.e. when years with total litter loss or absence of reproduction were removed) for the dataset used for the analyses in this manuscript, average recruitment (the number of female yearlings produced by a female for a given year) was 1.44 and 1.48 for the tactics 1.5-year and 2.5-year, respectively. We have tested whether recruitment (the number of yearling females) varied according to maternal care tactic using a generalized linear mixed effects model with recruitment as the response variable and tactic as the explanatory factor. We also included maternal identity as a random factor and used a Poisson distribution with the R package “lme4”. Recruitment did not vary with maternal care tactic ($\beta = 0.026$, $P=0.886$). Therefore, recruitment rates calculated in the present manuscript are not biased by potential differences in the number of female yearlings produced at each reproductive event. Reduced recruitment rate for females using the 2.5-year tactic are thus a reflection of a reduction in their number of reproductive opportunities. However, while there may be differences in litter size when investigated at the cub-of-the-year stage and when including also males, we have chosen not to include this analysis in the manuscript as this is currently part of an ongoing project investigating the determinants of the duration of maternal care and we wish to avoid potential overlap.*

There is a focus (‘crucial question’, line 125) on which phenotype yields the highest fitness. Pls consider modifying that discussion around the likelihood that the two phenotypes likely will (and have a history of) varying in their relative frequencies as the environment (resources, mortality regime) changes. Its actually an interesting result that (under the last 20 years) the two have similar fitness. This might be why the phenotypes have been maintained. Its an argument about the maintenance of biological (life history) diversity. You get there in lines 145 with discussion on ‘promoting...alternative maternal care tactics’, but it comes late as is overshadowed by the focus on ‘optimality’ (eg line 295).

Response: Following the reviewer's suggestion, we have modified the discussion to remove the focus on optimality and now discuss the possibility that the benefit of the two reproductive tactics may fluctuate over time (L252-259).

Line 83 and others – pls make clear that these are annual finite rates of mortality

Response: Recommendation followed (L98-104).

I would consider formally combining the Results and Discussion. You include lots of discussion matter before the 1-paragraph 'Discussion' section at the end.

Response: To comply with Nature Communications formatting requirements, a separate Discussion section has been added after the Results section.

Chris Darimont

Literature Cited:

- Bieber, C. & Ruf, T. (2005) Population dynamics in wild boar *Sus scrofa*: ecology, elasticity of growth rate and implications. *Journal of Applied Ecology*, **42**, 1203–1213.
- Caswell, H. (2001) *Matrix Population Models: Construction, Analysis, and Interpretation*, 2nd ed. Sinauer Associates, Sunderland.
- Gervasi, V., Nilsen, E.B., Sand, H., Panzacchi, M., Rauset, G.R., Pedersen, H.C., Kindberg, J., Wabakken, P., Zimmermann, B., Odden, J., Liberg, O., Swenson, J.E. & Linnell, J.D.C. (2012) Predicting the potential demographic impact of predators on their prey: A comparative analysis of two carnivore-ungulate systems in Scandinavia. *Journal of Animal Ecology*, **81**, 443–454.
- Gosselin, J., Zedrosser, A., Swenson, J.E. & Pelletier, F. (2015) The relative importance of direct and indirect effects of hunting mortality on the population dynamics of brown bears. *Proceedings of the Royal Society B*, **282**, 20141840.
- Hertel, A.G., Bischof, R., Langval, O., Mysterud, A., Kindberg, J., Swenson, J.E. & Zedrosser, A. (2017) Berry production drives bottom-up effects on body mass and reproductive success in an omnivore. *Oikos*, 10.1111/oik.04515.
- Newman, K.B., Buckland, S.T., Morgan, B.J.T., King, R., Borchers, D.L., Cole, D.J., Besbeas, P., Gimenez, O. & Thomas, L. (2014) *Modelling Population Dynamics* (eds AP Robinson, ST Buckland, P Reich, and M McCarthy). Springer, New York.
- Otto, S. & Day, T. (2007) *A Biologist's Guide to Mathematical Modeling in Ecology and Evolution*. Princeton University Press, New Jersey.
- Pelletier, F., Moyes, K., Clutton-brock, T.H. & Coulson, T. (2011) Decomposing variation in population growth into contributions from environment and phenotypes in an age-structured population. *Proceedings of the Royal Society B*, doi:10.1098/rspb.2011.0827.
- Saether, B.-E. & Bakke, O. (2000) *Avian Life History Variation and Contribution of*

- Demographic Traits To the Population Growth Rate. Ecology, 81, 642–653.*
- Sibly, R.M. & Hone, J. (2002) *Population growth rate and its determinants: an overview. Philosophical transactions of the Royal Society of London B, 357, 1153–1170.*
- Stenseth, N.E., Lutnæs, P.N., Bjarnadóttir, V., Dahle, B., Fossum, K.H., Jørgensen, P., Johansen, T., Neumann, W., Opseth, O., Rønning, O., Steyaert, S.M.J.G., Zedrosser, A., Brunberg, S. & Swenson, J.E. (2016) *Seasonal and annual variation in the diet of brown bears Ursus arctos in the boreal forest of southcentral Sweden. Wildlife Biology, 22, 107–116.*
- Swenson, J.E., Dahle, B. & Sandegren, F. (2001a) *Intraspecific predation in Scandinavian brown bears older than cubs-of-the-year. Ursus, 12, 81–91.*
- Swenson, J.E., Sandegren, F., Brunberg, S. & Segerström, P. (2001b) *Factors associated with loss of brown bear cubs in Sweden. Ursus, 12, 69–80.*
- Zedrosser, A., Pelletier, F., Bischof, R., Festa-Bianchet, M. & Swenson, J.E. (2013) *Determinants of lifetime reproduction in female brown bears: early body mass, longevity, and hunting regulations. Ecology, 94, 231–240.*

REVIEWERS' COMMENTS:

Reviewer #2 (Remarks to the Author):

The authors have done a very thorough job at addressing all comments, even providing new analyses

The revised manuscript is improved and I believe ready for publication

Reviewer #3 (Remarks to the Author):

In my view, the authors have done an excellent job in responding effectively and comprehensively to the reviewers' comments. The only concern I would raise relates to the small sample sizes in Figure 1, on which the proportional estimates of the two tactics (1.5 and 2.5 yr) are empirically based (assuming I have interpreted Figure 1 correctly).

Much is made of the observation that the 2.5-year tactic was not evident prior to 1995. Yet the average number of litters examined on an annual basis from 1987 to 1994 was only 2.5. Sample sizes increased somewhat from 1999 to 2011, but they remain relatively small.

At a minimum, I recommend that the authors acknowledge this issue in the Discussion. I like the manuscript a great deal. The combination of long-term empirical data with life-history modelling is very nice to see. It would be unfortunate if the importance of the authors' work was to be potentially diminished because of the small sample sizes referred to above.

Reviewers' comments:

Reviewer #2 (Remarks to the Author):

The authors have done a very thorough job at addressing all comments, even providing new analyses

The revised manuscript is improved and I believe ready for publication

Response: We are thankful to the reviewer for these positive comments.

Reviewer #3 (Remarks to the Author):

In my view, the authors have done an excellent job in responding effectively and comprehensively to the reviewers' comments. The only concern I would raise relates to the small sample sizes in Figure 1, on which the proportional estimates of the two tactics (1.5 and 2.5 yr) are empirically based (assuming I have interpreted Figure 1 correctly).

Much is made of the observation that the 2.5-year tactic was not evident prior to 1995. Yet the average number of litters examined on an annual basis from 1987 to 1994 was only 2.5. Sample sizes increased somewhat from 1999 to 2011, but they remain relatively small.

At a minimum, I recommend that the authors acknowledge this issue in the Discussion. I like the manuscript a great deal. The combination of long-term empirical data with life-history modelling is very nice to see. It would be unfortunate if the importance of the authors' work was to be potentially diminished because of the small sample sizes referred to above.

Response: The reviewer interpreted Figure 1 correctly. Figure 1 shows, using empirical data, the observed proportion of litters that have received 1.5 years or 2.5 years of maternal care in the study population from 1987 to 2015. The aim was to show that the proportion of litters that have received 2.5 years of maternal care is higher in recent years compared to in earlier years. Following a suggestion from the editor, we now provide a statistical test to show that the odds that a litter receive 2.5 years of maternal care increased in the study population over time (L92-94 and L338-344). Moreover, we acknowledge the difference in sample size between early and late periods of the study in the discussion and provide a comparison of the percentage of each tactic for 2 periods where a larger and similar number of litters had been monitored (1987:2004, n=80, 2005:2015; n=84) (L234-240).